# *Candida glabrata* Empyema Thoracis—A Post-COVID-19 Complication

**DOI:** 10.3390/jof8090923

**Published:** 2022-08-30

**Authors:** Neeraja Swaminathan, Katherine Anderson, Joshua D. Nosanchuk, Matthew J. Akiyama

**Affiliations:** 1Department of Medicine, Division of Infectious Diseases, Albert Einstein College of Medicine and Montefiore Medical Center, Bronx, NY 10467, USA; 2Case Western Reserve University School of Medicine, Cleveland, OH 44106, USA

**Keywords:** COVID-19, *Candida glabrata*, *Candida* empyema

## Abstract

The COVID-19 pandemic is associated with a significant increase in the incidence of invasive mycosis, including pulmonary aspergillosis, mucormycosis, and candidiasis. Fungal empyema thoracis (FET) is an uncommon clinical presentation of invasive fungal disease (IFD) associated with significant mortality. Here, we describe the first report of a patient with post-COVID-19 multifocal necrotizing pneumonia complicated by a polymicrobial empyema that included *Candida glabrata. Candida* empyemas represent another manifestation of a COVID-19-associated fungal opportunistic infection, and this infrequently encountered entity requires a high degree of clinical suspicion for timely identification and management. Therapy for empyemas and other non-bloodstream *Candida* infections may differ from candidemia due to several pharmacokinetic parameters impacting bioavailability of the antifungal in the affected tissue (e.g., pleural space) and is an area that needs more investigation.

## 1. Introduction

Invasive candidiasis, especially candidemia, has increasingly been recognized as a complication of COVID-19 and is associated with an estimated mortality of 19–40% overall, with as high as 70% in critically ill patients [1]. *Candida* empyemas are a rare but severe manifestation of invasive candidiasis with scarce data on their optimal management [2]. Additionally, the landscape of *Candida* infections continues to shift with a growing number of non-*albicans Candida* (NAC) cases being reported. This is clinically relevant because of both the intrinsic and acquired resistance to antifungals seen in these organisms [3]. The goals of this case report are to review our current knowledge of *Candida* empyemas pertaining to risk factors, such as COVID-19, and to discuss the pathogenesis and management of this uncommon mycoses.

## 2. Case Presentation

A female in her 60s with poorly controlled diabetes mellitus (hemoglobin A1c > 14 g/dL) presented 10 days after her discharge from another facility where she was diagnosed with COVID-19. She was incompletely vaccinated. During the initial admission, she was hypoxic and required oxygen supplementation via nasal cannula. She was treated with a 5-day course of remdesivir (200 mg on the first day, followed by 100 mg daily) and 10 days of dexamethasone (6 mg daily) and was discharged home after 7 days on two liters of supplemental oxygen via nasal canula. The patient noted a worsening dry cough, dyspnea, and fatigue after her discharge and endorsed a loss in appetite and a loss in weight. She denied difficulty or pain while swallowing, nausea, vomiting, abdominal pain, and diarrhea. She had no significant travel or exposure to sick contacts and no substance use or underlying lung disease. Her home medications included only insulin and metformin for diabetes.

When she came to our center, she appeared chronically ill, was tachycardic (108 beats per minute), and required 4 L of oxygen via nasal canula to achieve oxygen saturation > 95%. Upon respiratory exam, she had coarse rhonchi bilaterally with reduced breath sounds in the left lower zone. She had mild anasarca and pitting pedal edema. The remainder of her cardiovascular, abdominal, and neurological examination was unremarkable.

## 3. Investigations

In our emergency department, a chest radiograph showed bilateral air space opacities and a left-sided loculated effusion in the lower zone. A follow-up computed tomography (CT) demonstrated moderate left pleural effusion and bilateral patchy airspace opacities with areas of ground-glass attenuation and scattered nodules in the bilateral lung fields (Figure 1). There was no intra-abdominal pathology noted, and the stomach, pancreas, and esophagus were all within normal limits.

The admission labs were notable for hyperglycemia to 500 mg/dL, but no ketoacidosis. Albumin was low at 2 g/dL, and the rest of the electrolytes and liver function tests were within normal ranges. Her leukocyte count was 6400 cells/cubic mm (89% neutrophils). Her hemoglobin was 9 g/dL, and the platelet count was 433,000 cells/cubic mm.

HIV testing was negative, as was a nasopharyngeal swab for SARS-CoV-2, influenza, and respiratory syncytial virus (RSV). Urine antigen tests for *Streptococcus pneumoniae* and *Leginonella pneumophila*, as well as methicillin-resistant *Staphylococcus aureus* (MRSA) testing of the nares, were negative.

The patient had a thoracentesis at admission with an expression of frank pus, followed by the placement of a chest tube. Her pleural fluid leukocyte count was 709,520 cells/cubic mm (85% neutrophils), with lactate dehydrogenase (LDH) of 8440 U/L, glucose of 13 mg/dL, and protein of 1.6 g/dL. The serum LDH was 198 U/L (upper limit 240 U/L), glucose was 335 mg/dL, and protein was 4.5 g/dL.

Pleural fluid Gram staining revealed Gram-positive cocci (GPC) in pairs, Gram negative bacilli (GNB), and yeast with a morphology consistent with *Candida*. A subsequent serum cryptococcal antigen was negative. Blood cultures were negative.

The pleural fluid cultures grew extended-spectrum beta lactamase (ESBL), producing *Klebsiella pneumoniae* and *Enterococcus faecalis,* which was sensitive to ampicillin and vancomycin. The yeast noted in the culture was initially identified as *Candida firmetaria* by our microbiology laboratory based on an automated biochemical method (BD Phoenix). However, analysis by matrix-assisted laser desorption/ionization time-of-flight mass spectrometry (MALDI-TOF MS) identified the isolate as *Candida glabrata.* The minimum inhibitory concentrations (MICs) for this organism are summarized in Table 1, as follows.

*C. glabrata* is susceptible to fluconazole, voriconazole, micafungin, and amphotericin B according to the Clinical and Laboratory Standards Institute (CLSI) criteria.

## 4. Differential Diagnosis

The patient presented with a subacute course of respiratory and constitutional symptoms 2 weeks after an initial diagnosis of COVID-19. Her physical examination was consistent with a consolidation in the lower left posterior lung. A radiographic examination revealed the presence of a multifocal necrotizing pneumonia with a left-sided empyema thoracis. An analysis of the pleural fluid revealed a polymicrobial infection, including ESBL *K.*
*pneumoniae*, *E. faecalis*, and *C. glabrata*.

## 5. Treatment

The patient was initially started on empiric intravenous (IV) vancomycin at 1 g every 24 h and piperacillin-tazobactam at 4.5 g every 8 h. IV micafungin at 100 mg every 24 h was added on day two once the pleural fluid Gram stain was reviewed and the yeasts were identified. On day three of admission, IV meropenem at 500 mg every 6 h was started in place of piperacillin-tazobactam in response to the *K. pneumoniae* susceptibilities. After 4 days of micafungin, the patient was switched to an oral voriconazole tablet of 300 mg every 12 h for two loading doses, followed by 200 mg every 12 h for enhanced drug bioavailability in the pleural space. Voriconazole was empirically selected, as *C. glabrata* are generally susceptible to this azole, even in the setting of fluconazole resistance. The patient’s respiratory symptoms improved on this antimicrobial regimen.

## 6. Outcome and Follow-Up

Unfortunately, after her initial clinical improvement, the patient developed an acute cardiac arrest on day ten of admission. Despite resuscitative efforts, the patient expired. The most likely cause of death was thought to be an arrythmia or a sudden hemothorax or pneumothorax. The patient’s family declined an autopsy. Figure 2 summarizes the clinical course of this patient.

## 7. Discussion

The incidence of pleural effusions in the setting of COVID-19 ranges from 5–9%, but fungal etiologies for these empyemas are exceedingly rare [4]. To date, the literature available regarding all conditions resulting in *Candida* empyemas is limited only to case reports and case series. The Infectious Diseases Society of America (IDSA) guidelines for the management of candidiasis do not specifically address *Candida* empyemas [5]. Australian treatment guidelines for invasive candidiasis in hematology-oncology and intensive care unit (ICU) settings note that, for *Candida* empyemas, fluconazole is first-line agent if the isolate is susceptible and mention echinocandins, voriconazole, and posaconazole as options if there is concern for fluconazole resistance. The recommended treatment duration in these guidelines is a minimum of 2 weeks. However, these are level III recommendations [6].

The identified risk factors for the development of *Candida* empyemas include elderly age, underlying comorbidities (such as diabetes mellitus, chronic kidney disease, cirrhosis, and malignancies), septic shock, receipt of steroids or chemotherapy, prolonged antibiotic use, ICU stay, and intra-abdominal diseases (such as esophageal rupture, complicated surgery, pancreatitis, etc.) [2,7,8,9]. The distribution of *Candida* species in large case series describing the entity of fungal empyemas has shown that a majority are secondary to *C. albicans*, while NAC are less common [2,7,8]. However, the ongoing shift in prevalence of NAC may alter the epidemiology of this condition in the future. Concomitant bacterial empyemas are frequently present in patients with fungal empyemas. For example, in a case series from Taiwan with 63 patients, 49% had a bacterial infection, and 14% also had fungemia [8]. Mortality rates are variable in these case series, ranging from 27 to 73% [2,7,8,9]. It appears that pleural drainage, particularly in video-assisted thoracoscopic surgery (VATS), reduced the risk of mortality. In addition, higher mortality is seen in older studies, whereas newer case series have improved outcomes, which could reflect improvements in overall ICU care [2].

There are no studies that compare antifungals in empyema to demonstrate the superiority of one agent. This is unlike candidemia, where echinocandins are recommended as empiric first-line management [5]. In a retrospective case series of eighty-one patients with fungal empyemas at two tertiary care centers, echinocandins were identified as an independent predictor of increased risk for mortality [2]. While the exact reasons for azoles being superior to echinocandins were not clear, the authors attributed the outcomes to the antifungal pharmacokinetics of azoles being superior to echinocandins in achieving adequate levels in pleural fluid. The factors that influence drug concentration include pH, oxygen concentration, protein binding, degree of pleural inflammation, size of the effusion, and presence of loculations [10]. In a study that looked at anidulafungin in the management of *Candida* empyemas, the ratio of the area under the concentration (AUC) to the time curve between pleural fluid and plasma was only 12.5% [11]. Voriconazole and fluconazole were noted to have good penetration into the pleural fluid, producing trough concentrations similar to paired plasma concentrations [2,12]. In contrast, echinocandin concentrations within infected pleural fluid were between 9–15% and 57–67% of those in plasma [2,10]. However, the evidence at present remains insufficient to make any definitive recommendation favoring azoles in the management of fungal empyemas. There are also scant data on the utility of pleural irrigation, although an isolated case report documented the instillation of antifungals [9]. Amphotericin B also reportedly had a poor pleural-to-serum concentration ratio [10]. Apart from these pharmacokinetic parameters, another important consideration in the choice of antifungals is that of antimicrobial resistance. Several NAC species are associated with resistance to azoles, especially fluconazole, but these isolates are often susceptible to newer azoles, such as voriconazole, Posaconazole, and isavuconazole [3].

Another aspect of this case that merits attention is the challenges associated with the accurate identification of *Candida* species—especially NAC species. Although colony color and morphology on a chromogenic medium (CHROMagar) allow the ready differentiation of *C. albicans* and NAC species, distinguishing specific NAC species is more difficult [13,14]. Identification panels such as BD Phoenix, Vitek, AuxaColor, etc. are miniaturized biochemical-based tests that contain a series of conventional, chromogenic, and fluorogenic tests, as well as growth- and enzymatic-based substrates. The microbial utilization and degradation of substrates are detected by chromogenic and fluorogenic indicators. The results are compared against a standard. However, not all organisms of a particular strain are 100% uniform with respect to some of the results of fermentation, oxidation, hydrolysis. Moreover, these tests may also not be able to distinguish closely related organisms that may produce similar results [15,16]. Thus, there is a small chance of erroneous identification, as encountered in our patient. MALDI-TOF MS is a rapid and more reliable means of definitive *Candida* species identification [17,18]. *C. firmetaria*, formerly known as *C. lambica,* which was the initial identification by our laboratory, is a rare pathogen in humans with only a handful of case reports noted in the literature. It bears morphological similarity to *C. kruseii* [19,20,21]. *C. glabrata* is a more common variety of NAC and has previously been described as a cause of *Candida* empyemas in patients without COVID-19 [2,7,8,9].

The fungal infections commonly described in the setting of COVID-19 include COVID-19-associated pulmonary aspergillosis (CAPA), COVID-associated mucormycosis (CAM), and COVID-associated candidiasis (CAC). CAPA has a highly variable cumulative incidence in the ICU (from 1 to 40%) and an attributable mortality of around 33%, and these cases have predominantly been due to *Aspergillus fumigatus* [22,23]. In the case of CAM, while the exact incidence is not known, it is close to 7 per 1000 COVID-19 cases, which is over 50 times higher than the prior highest incidence data available. There has been a clear rise in cases globally during the pandemic, especially in developing countries, such as India, which already have a high baseline prevalence of mucormycosis. Most of these cases have been noted in the diabetic population, and the mortality rates have ranged from 25 to 50% [24].

COVID-19 as a risk factor for invasive candidiasis has only recently emerged. A meta-analysis looking at COVID-19-associated candidiasis (CAC) found that the pooled prevalence of resistant NAC, specifically *Candida auris*, was 5.7% but with a mortality as high as 67% [25]. CAC can be due to two major groups of risk factors. The first group includes generic risk factors such as age, premorbid comorbidities, critical illness with artificial lines and tubes, parenteral nutrition, etc. The second group comprises more COVID-19-specific risks, which include the use of extracorporeal membrane oxygenation (ECMO), the administration of corticosteroids, and notably, COVID-19-mediated damage to the lung epithelium. There are also reports of an association between the use of Interleukin- 6 inhibitors, such as tocilizumab, and the development of candidiasis [1]. *Candida* colonization of the airway is also common in critically ill patients. One study looked at the characterization of *Candida* colonization and COVID-19, and polymerase chain reaction (PCR) sequencing demonstrated that sixty-nine out of one hundred patients had *Candida* species in bronchoalveolar lavage fluid (BAL) specimens. While *C. albicans* colonization had no discernable impact on COVID-19 severity, *C. glabrata* colonization did [26]. Moreover, respiratory *Candida* colonization was present in over one-third of patients with *Candida* empyemas within 4 weeks of disease manifestation [7]. The growth of *Candida* in BAL and sputum cultures is usually deemed to be from oral colonizers and, therefore, *Candida* pneumonitis is often untreated [26]. However, when *Candida* grows in blood cultures or pleural cultures, it cannot be disregarded as a colonizer or contaminant.

COVID-19-related *Candida* empyemas have been infrequently reported in the literature, and we identified three other cases, each with *C. albicans*. Table 2 summarizes the patient characteristics.

Our patient is the first description of empyema thoracis due to *C. glabrata* following COVID-19. Her risk factors included age, uncontrolled diabetes mellitus, and a receipt of steroids for COVID-19 treatment. She presented with a post-COVID-19 necrotizing bacterial pneumonia and pleural effusion that became super-infected with *Candida*. We did not have a prior sputum culture on this patient to assess if she had known colonization with Candida. There was also no documentation of her exposure to any antimicrobials, except remdesivir, prior to her admission.

Our patient underscores the importance of considering fungal empyema in COVID-19 patients who present with complicated pneumonias, as this disease requires prompt pleural drainage and antifungal treatment.

## 8. Conclusions

*Candida* empyema thoracis is an invasive candidiasis that frequently occurs in the absence of candidemia;Fungal empyema thoracis commonly presents as a polymicrobial infection, predominantly with a concomitant bacterial infection;COVID-19 is a newly identified risk factor for fungal empyema;The management of *Candida* empyemas involves prompt pleural drainage and systemic antifungals;Optimum treatment for non-bloodstream candidiasis may vary in important ways from the management of candidemia, and there are pharmacologic reasons for predicting that azoles are superior to echinocandins in the management of *Candida* empyemas.

## Figures and Tables

**Figure 1 jof-08-00923-f001:**
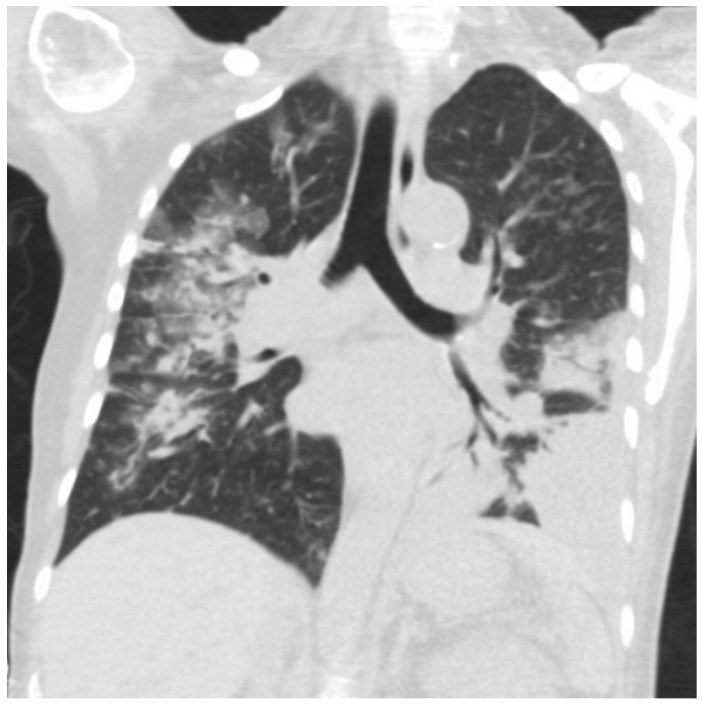
CT of thorax showing left-sided pleural effusion and bilateral airspace opacities.

**Figure 2 jof-08-00923-f002:**
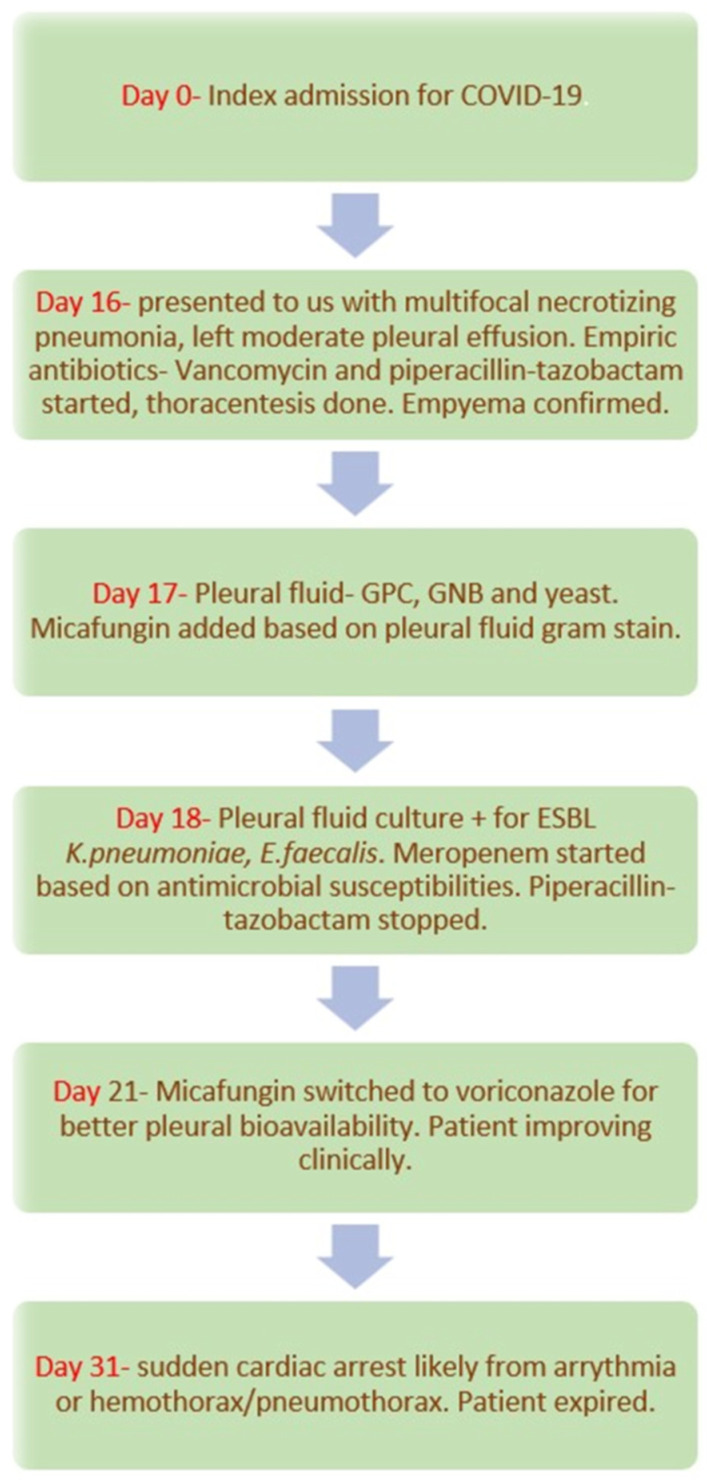
Timeline of patient’s clinical course.

**Table 1 jof-08-00923-t001:** MICs for the isolated *C. glabrata*.

DRUG	MIC
Voriconazole	0.12 ug/mL
Anidulafungin	0.06 ug/mL
Caspofungin	0.06 ug/mL
Fluconazole	4 ug/mL
Itraconazole	0.5 ug/mL
Isavuconazole	0.12 ug/mL
Posaconazole	0.5 ug/mL
Micafungin	0.015 ug/mL
Amphotericin B (E-test)	0.19 ug/mL
5-Fluorocytosine (E-test)	0.016 ug/mL

**Table 2 jof-08-00923-t002:** Three prior reports of *C. albicans* empyemas associated with COVID-19.

Study	Age, Gender	Comorbid Conditions	Fungus Isolated	COVID-19 Management	Empyema Treatment	Concomitant Bacterial Infection	Outcome
Sharma et al. [27]	55, male	Hypertension	*C. albicans*	Not mentioned	Tube thoracostomy, micafungin	MRSA in respiratory culture	Unclear: still admitted at the time of publication
Qasem et al. [28]	52, male	None	*C. albicans*	ECMO	Chest tube, decortication	-	Expired
Glendening et al. [29]	73, male	Congestive heart failure	*C. albicans*	Hydroxychloroquine intubated	Chest tube, fluconazole	*Moraxella* bacteremia	Discharged

## Data Availability

Not applicable.

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
