# Peer review of "Candida glabrata Empyema Thoracis—A Post-COVID-19 Complication"

_jof, 2022, doi:10.3390/jof8090923_

Round 1

Reviewer 1 Report

Dear Editor,

Thank you very much for the opportunity to review this manuscript. In it, the authors describe a case of COVID-19 associated Candida infection (CAC) due to Candida glabrata. The report is easy to read and well written, congratulations! However, there are certain minor aspects I would suggest to improve.

ABSTRACT

- Here you mention COVID-19 associated pulmonary aspergillos (CAPA) and mucormycosis (CAM), but then in the main text I cannot find any mention to such infections. I would suggest to include one sentence. Suggested references: Salmanton-Garcia et al Emerg Infect Dis 2021, Koehler et at Eur Respir J 2022, Werthman-Ehrenreich A Am J Emerg Med 2021

TEXT

- Additionally, when mentioning CAC, you could also mention the one due to C. auris, or previous reports on C. glabrata, or CAC with resistances

- I would suggest to include a figure (similar to a Gant chart) where all the relevant information (treatment, diagnosis, admission) is placed, starting from Day 0 since hospital admission, or COVID-19 diagnosis. This is very helpful to better understand the evolution of the patient, as all events are described with their start and stop days

- I would suggest to include the MIC values for the tested antifungals

- Drug dosages would be welcome to be included in the text. Also the drug formulation, if applicable

- "The IDSA guidelines for the management of candidiasis does not specifically address Candida empyema". What about other guidelines?

- Please, use the term MALDI-TOF MS

- Please, verify and confirm that the meaning of each abbreviation/acronym is explained at the first time such term is mentioned

Author Response

Dear Reviewer,

At the outset, we would like to convey our gratitude for your valuable input. Please find below our response/revisions based on your feedback.

1.You mention COVID-19 associated pulmonary aspergillosis (CAPA) and mucormycosis (CAM), but then in the main text I cannot find any mention to such infections. I would suggest to include one sentence.

Response- Thank you for this input, We have added lines 184-193 to discuss very briefly about CAPA and CAM

2. Additionally, when mentioning CAC, you could also mention the one due to C. auris, or previous reports on C. glabrata, or CAC with resistances
Response- We have included this in lines 196-197

3. I would suggest to include a figure (similar to a Gant chart) where all the relevant information (treatment, diagnosis, admission) is placed, starting from Day 0 since hospital admission, or COVID-19 diagnosis. This is very helpful to better understand the evolution of the patient, as all events are described with their start and stop days
Response- That was a very helpful tip and we have inserted a graphic - Figure 2. It is not exactly a Gantt chart but its a flowchart which sequentially notes the key events from index admission to eventual death.

4. I would suggest to include the MIC values for the tested antifungals

Response- We have included this in table 1. 

5. Drug dosages would be welcome to be included in the text. Also the drug formulation, if applicable

Response- I have included the same in the treatment section

6. The IDSA guidelines for the management of candidiasis does not specifically address Candida empyema". What about other guidelines?

Response- After reviewing other guidelines like ESCMID, ESICM, Regional guidelines from Taiwan/Middle East/Canada and Australia, we only found one Australian guideline that specifically mentions some guidance for Candida empyema. This is referenced in lines 121-126, Reference 6. 

7. Please, use the term MALDI-TOF MS

Response- We have changed all the abbreviations from MALDI to MALDI-TOF MS

8. Please, verify and confirm that the meaning of each abbreviation/acronym is explained at the first time such term is mentioned

Response- Yes, we have ensured that all abbreviations/acronyms are explained the first time they appear in text. 

Reviewer 2 Report

The manuscript presents a report study on a case of Candida empyema thoracis caused by C. glabrata in a post COVID-19 diabetic patient. The case was well presented, however some editorial issues should be revised.

Line 36: add space between "6" and "mg". Similar oversights are present in the "Investigations" part. Additionally, in line 64 "cells/cubic" is missing.

Line 56: replace square with round brackets.

Line 116: [2,6-8] instead of "[2,6,7,8]". Similarly in lines 122-123 and 167. Additionally, [18-20] in lines 165-66.

Line 141: 9–15% and 57–67%, instead of "9%–15% and 57%–67%"

Author Response

Dear Reviewer

I would like to convey on behalf of all authors our sincere gratitude for your feedback. You had suggested some minor revisions as follows

Line 36: add space between "6" and "mg". Similar oversights are present in the "Investigations" part. Additionally, in line 64 "cells/cubic" is missing.

Line 56: replace square with round brackets.

Line 116: [2,6-8] instead of "[2,6,7,8]". Similarly in lines 122-123 and 167. Additionally, [18-20] in lines 165-66.

Line 141: 9–15% and 57–67%, instead of "9%–15% and 57%–67%"

Response - All of the above editing issues have been addressed.

Regards

Neeraja Swaminathan